# Magnitude and Extent of *Helicoverpa zea* Resistance Levels to Cry1Ac and Cry2Ab2 across the Southeastern USA

**DOI:** 10.3390/insects14030262

**Published:** 2023-03-07

**Authors:** Dominic Reisig, G. David Buntin, Jeremy K. Greene, Silvana V. Paula-Moraes, Francis Reay-Jones, Phillip Roberts, Ron Smith, Sally V. Taylor

**Affiliations:** 1Department of Entomology, The Vernon James Center, North Carolina State University, Plymouth, NC 27962, USA; 2Department of Entomology, University of Georgia, Tifton, GA 31793, USA; 3Department of Plant and Environmental Sciences, Clemson University, Blackville, SC 29817, USA; 4Entomology and Nematology Department, University of Florida, Jay, FL 32565, USA; 5Department of Entomology and Plant Pathology, Auburn University, Auburn, AL 36849, USA; 6Department of Entomology, Virginia Polytechnic Institute and State University, Suffolk, VA 23437, USA

**Keywords:** bioassays, Bt cotton, diet overlay, resistance

## Abstract

**Simple Summary:**

After resistance is first detected, continued resistance monitoring to describe the magnitude and extent of the resistance can inform pest management decisions on how to effectively manage the resistant populations. We collected *Helicoverpa zea* larvae from various plant hosts across the southeastern USA, and compared their survival to susceptible populations on a diet overlaid with the Bt toxins Cry1Ac and Cry2Ab2 for resistance estimates. Surprisingly, 62.5% of the tested populations were resistant to Cry2Ab, whereas only 37.5% of them were resistant to Cry1Ac. This contrasts with estimates in both the mid-southern and southeastern USA, where Cry1Ac, Cry1A.105, and Cry2Ab2 resistance increased over time and was found in a majority of populations. Both Cry1Ac and Cry2Ab resistance ratios were variable, but Cry2Ab resistance ratios were significantly higher than Cry1Ac resistance ratios in most of the tested populations. This indicates that cotton expressing Cry proteins in the southeastern USA was at variable risk for damage, in contrast to previous findings of increased damage in Cry-expressing cotton over time in this region.

**Abstract:**

After resistance is first detected, continued resistance monitoring can inform decisions on how to effectively manage resistant populations. We monitored for resistance to Cry1Ac (2018 and 2019) and Cry2Ab2 (2019) from southeastern USA populations of *Helicoverpa zea*. We collected larvae from various plant hosts, sib-mated the adults, and tested neonates using diet-overlay bioassays and compared them to susceptible populations for resistance estimates. We also compared LC_50_ values with larval survival, weight and larval inhibition at the highest dose tested using regression, and found that LC_50_ values were negatively correlated with survival for both proteins. Finally, we compared resistance rations between Cry1Ac and Cry2Ab2 during 2019. Some populations were resistant to Cry1Ac, and most were resistant to CryAb2; Cry1Ac resistance ratios were lower than Cry2Ab2 during 2019. Survival was positively correlated with larval weight inhibition for Cry2Ab. This contrasts with other studies in both the mid-southern and southeastern USA, where resistance to Cry1Ac, Cry1A.105, and Cry2Ab2 increased over time and was found in a majority of populations. This indicates that cotton expressing Cry proteins in the southeastern USA was at variable risk for damage in this region.

## 1. Introduction

When pest resistance to insecticides is documented, emphasis is usually placed on a single initial finding or a few limited occurrences. The rationale for this is sound as resistance monitoring is expensive, and changes in management with insecticides should be made as early as possible once resistance is detected, if not before. However, growers may be reluctant to make management changes if those changes are costly, difficult to implement or if effective alternatives are unavailable. Because insecticide susceptibility is a common pool resource [1], the actions of a single grower have little impact on pest population susceptibility relative to the collective action of many growers.

Resistance can be either field-evolved resistance, defined as a “genetically based decrease in susceptibility to a pesticide in a population caused by exposure to the pesticide in the field”, or practical resistance, defined as “field-evolved resistance that reduces pesticide efficacy and has practical consequences for pest control” [2]. In some cases, practical resistance may not be recognized as having practical consequences for pest management until sometime after initial discovery. For example, while practical resistance of *Helicoverpa zea* (Boddie), to Cry2Ab2 in cotton, *Gossypium hirsutum* (L.), was demonstrated in 2005 [3], yields of cotton expressing Cry2Ab2 peaked from 2010–2012, despite the declining efficacy of the pest [4]. Moreover, pest management changes for *H. zea* to cotton expressing Cry2Ab2 were not needed until 2016 [5,6].

Because there can be a considerable time lag between the initial discovery of resistance and the required changes in pest management, investigations beyond the initial discovery may be important. For example, *Bacillus thuringiensis* (Bt) resistance in *H. zea* is highly influenced by the landscape on both local and regional population levels. Early studies of *H. zea* moths suggested that regional host use differences could drive the susceptibility of *H. zea* to Bt proteins in the southeastern and mid-southern USA, which was supported by modeling [7,8,9]. Later studies confirmed that Bt cotton efficacy in the field varied across this region [4], as did the survival of *H. zea* in laboratory bioassays across the region [10]. In addition to anecdotal evidence that crop composition influences Bt resistance in *H. zea* [10], experimental evidence demonstrated that, at the local level (1–10 km radius) for *H. zea*, injury and damage to Bt crops, survival in Bt bioassays, and population densities of *H. zea* were highly regulated by the abundance of Bt and non-Bt crops in the local landscape during the previous and current year [11,12,13,14]. Therefore, the initial detection of practical resistance in one region or landscape may not have the same implications for pest management in another region, even for a pest like *H. zea* that is migratory and panmictic [15]. Moreover, some researchers have encouraged the documentation of the extent and magnitude of Bt resistance in the field to inform current and future resistance models [16].

Previous studies have documented the extent of *H. zea* resistance to Cry1Ac (expressed in cotton) across the southeastern USA during 2016 [5] and Cry1A.105 (expressed in corn, *Zea mays* L.), and Cry2Ab2 (expressed in some corn and cotton hybrids and varieties) in North Carolina and South Carolina during 2017 and 2018 [17]. The purpose of our study was to build on these investigations by evaluating the extent and magnitude of Cry1Ac resistance in *H. zea* across the southeastern USA during 2018 and 2019, and Cry2Ab2 resistance during 2019.

## 2. Materials and Methods

### 2.1. Larval Collections

We collected *H. zea* from a number of sources (Appendix A), including larvae from Bt and non-Bt hosts (*n* = 200–300 per collection) and a light trap (as individual previously mated adult females). We only made two collections from the light traps. These collections were from 2019 and we designated them as Washington Co., NC 1 and Washington Co., NC 2. For larval collections, we placed a single larva into a 30 mL plastic cup (Solo, Mason, MI) onto a pre-filled portion (~10 mL) of diet (to make 4 L of diet, we used: 4 L distilled water, 53 g agar, 126 g casein, 108 g wheat germ, 96 g sucrose, 36 g salt, 40 g cellulose, 20 mL 4 M KOH, 72 g Vanderzant vitamin mixture, 6.4 g sorbic acid, 6.4 g methyl paraben, 12.8 g ascorbic acid, 40 mg fumidil B, 480 g streptomycin, 8 mL 10% formaldehyde, and 6 mL wheat germ oil). We shipped or drove the larvae and the adults in a cooler to the laboratory following collection. We then placed the adult females in oviposition cages, provisioned with 10% sucrose water and an ovipositional substrate (cheesecloth) that was placed over 1.9 L round plastic container and secured with a lid that had the middle cut out. When the adults laid eggs, we removed ovipositional substrates daily and placed them into individually sealed plastic 473 mL deli containers with a cotton ball moistened with three drops of water. Larvae that eclosed on the ovipositional substrate and the collected larvae were reared to the pupal stage in a growth chamber (14:10 L:D, 27 °C:24 °C L:D, 60% RH; Percival Scientific Plant Growth Chamber Model E-75L1; Percival Scientific, Perry, IA, USA). We then placed pupae in the oviposition cages as described earlier for sib-mating.

### 2.2. Bioassay Procedures

We only performed bioassays if there were sufficient neonates that eclosed within a 24-h period (*n* = 256). As a result, we were not able to perform bioassays on all collections (Appendix A). For comparison to a known Cry-susceptible *H. zea* population, we used a colony collected in 2016 from Winnsboro, LA, USA [10] that we have maintained in culture since that time (designated as Cry-susceptible). We performed a bioassay on this colony during both 2018 and 2019. As another comparison, we purchased eggs (Benzon Research, Carlisle, PA, USA) and performed bioassays during both 2018 and 2019 (designated as Benzon).

Cry1Ac bioassays were performed during 2018 and 2019, while Cry2Ab2 assays were performed during 2019. We modified artificial diet (Southland Product, Inc., Lake Village, AR, USA) for *H. zea* by adding casein to a 1.6:1 protein to carbohydrate ratio [18]. We aliquoted 0.8 mL of this diet into each of 128 wells in trays (Bio-Assay Tray-128 Cells; Frontier Agricultural Sciences, Newark, DE, USA) and allowed it to cool in a laminar flow hood. We covered the diet with plastic wrap and refrigerated it prior to use. The concentration of Cry1Ac was 20% in freeze-dried MVPII powder, and the concentration of Cry2Ab2 was 4 mg g^−1^ of freeze-dried corn leaf powder. Before the bioassays, we removed the trays of diet from refrigeration to allow for acclimation to room temperature and aliquoted onto each diet-filled well 0.01, 0.0316, 0.1, 0.316, 1, 3.16, or 10 μg/cm^2^ of Cry1Ac dissolved in 0.1% agarose solution or 0.01, 0.0316, 0.1, 0.316, 1, 3.16, or 10 μg/cm^2^ of Cry2Ab2 leaf powder equivalent dissolved in 0.1% agarose solution. For each bioassay, we applied the doses into 64 individual wells (40 μL per well, which completely and evenly covered the diet) in the Cry1Ac assays and a non-Bt leaf powder-dissolved buffer applied to 64 individual wells (200 μL per well) in the Cry2Ab2 assays. Buffer was applied to 64 individual wells (200 μL per well) as a check for each assay. This provided four replicates of 16 larvae each per concentration. We then placed the bioassay trays in a laminar flow hood until dry. Bayer Crop Science (St. Louis, MO, USA) provided Cry1Ac and Cry2Ab2, as described in [10].

Upon drying, we placed individual neonates onto wells using a fine-tipped paintbrush. Care was taken not to touch the diet to avoid cross contamination. We covered the wells using vented lids (Bio-Assay Tray Lid-16 Cells; Frontier Agricultural Sciences). After seven days, we removed the lids and prodded the larvae using a paintbrush. If the larvae did not move, we did not remove them. We weighed the larvae that moved and noted the instar. We considered larvae that did not move or molt into the second instar as functionally dead (molting inhibitory concentration), following [10,19]. We then averaged survival, weight, and instar for each concentration in a given bioassay. To calculate percentage larval inhibition for weight and percentage larval inhibition for instars, we took the average weight and instar number from the survivors at the highest toxin combination we tested, and divided this number from the average weight and instar number from the survivors from the control in each bioassay. We then multiplied this proportion by 100 [5,20].

### 2.3. Statistical Analyses

We used probit analysis PROC PROBIT [21] to calculate LC_50_ values (mortality defined as larvae that did not move when prodded with a paintbrush, or molt to the second instar). We report the Wald Chi square test statistic and the *p*-value from the analysis. In some cases, the LC_50_ was estimated to be higher than the highest concentration that we tested. We reported these values as greater than the highest concentration we tested in these cases. We calculated resistance ratios using the LC_50_ value for the Cry-susceptible colony within a given year for Cry1Ac as the denominator, and the LC_50_ value of individual bioassays as the numerator. Because the health of the Cry-susceptible colony was variable during 2018, and despite three tests with Cry2Ab2 leaf powder ending in variable or excessive mortality in the control wells or across doses, we did not report these results; rather, we calculated resistance ratios using the LC_50_ value for the Benzon colony for Cry2Ab2.

We also regressed LC_50_ values against the estimated slope from the probit analysis and percentage mortality, larval inhibition for weight, and larval inhibition for instar at the highest dose (10 μg/cm^2^) using PROC REG [21]. We performed separate regressions for each variable for Cry1Ac (2018 and 2019 combined) and Cry2Ab2 (2019). In cases where the LC_50_ was estimated higher than the highest concentration that we tested, we used 10 μg/cm^2^ as the LC_50_ value, even though this was an underestimate. We made transformations as needed to meet model assumptions. We reported R^2^ values and regression equations only when the regression model was significant (*p* < 0.05).

Finally, we compared the resistance ratios between Cry1Ac and Cry2Ab2 relative to the Benzon colony during 2019 using a paired *t*-test, the PROC TTEST [21]. For populations where the LC_50_ was estimated higher than the highest concentration that we tested, we conservatively used the resistance ratio calculated based on this highest concentration. Finally, we transformed the Cry2Ab2 resistance ratios by taking the square root of the values prior to the test to satisfy the assumption of normality. We considered the test significant if *p* < 0.05.

## 3. Results

During 2018, for Cry1Ac, the LC_50_ value of the susceptible colony was 0.33 μg/cm^2^, and the LC_50_ value of the Benzon colony was 0.32 μg/cm^2^ (Table 1). Five collections were tested for Cry1Ac resistance during this year, with resistance ratios ranging from >1 to over 30.

During 2019, for Cry1Ac, the LC_50_ value of the susceptible colony was 0.55 μg/cm^2^, and the LC_50_ value of the Benzon colony was 0.27 μg/cm^2^ (Table 2). Resistance ratios were based on that of the susceptible colony and ranged from <1 (Surry Co., NC) to >18 (Henry Co., AL and Santa Rosa Co., FL).

During 2019, for Cry2Ab2, the LC_50_ value of the Benzon colony was 0.07 μg/cm^2^ (Table 3). Resistance ratios were based on that of the Benzon colony and ranged from 2 (Florence, SC) to over >143 (Santa Rosa Co., FL 1).

Estimated slopes from the probit analysis were not related to LC_50_ values for either Cry1Ac (*F* = 0.28; 1, 18; *p* = 0.6054) or Cry2Ab2 (*F* = 1.46; 1, 11; *p* = 0.2521). Percentage mortality values at the highest dose (10 μg/cm^2^) showed a significant negative linear relation to LC_50_ values for both Cry1Ac (*F* = 56.10; 1, 18; *p* < 0.0001; R^2^ = 0.76; Percentage mortality values = 76.43 − (1.69 × LC_50_) and Cry2Ab2 (*F* = 5.25; 1, 11; *p* = 0.0427; R^2^ = 0.32; Percentage mortality values = 95.73 − (5.61 × LC_50_)). Larval inhibition values for weight were not related to LC_50_ values for Cry1Ac (*F* = 0.44; 1, 13; *p* = 0.5181). In contrast, larval inhibition values for weight showed a significant positive logarithmical relation to LC_50_ values for Cry2Ab2 (*F* = 8.37; 1, 10; *p* = 0.0160; R^2^ = 0.46; weight = 91.73 + *ln*(1.71 × LC_50_)). Larval inhibition values for instar were not related to LC_50_ values for either Cry1Ac (*F* = 2.91; 1, 10; *p* = 0.1188) or Cry2Ab2 (*F* = 0.10; 1, 14; *p* = 0.7608). Finally, Cry1Ac resistance ratios were lower than Cry2Ab2 (15 and 53, respectively; *t* = 5.26, *p* = 0.0003) among 2019 populations.

## 4. Discussion

Our study documents the variability of Cry resistance in *H. zea* across the southeastern USA. Using a resistance ratio of 10 as a cutoff value [10,22,23], some of the populations that we tested were resistant to Cry1Ac (six out of 16) and most were also resistant to CryAb2 (nine out of 12). This contrasts with estimates in the mid-south USA, where Cry1Ac and Cry2Ab2 resistance increased over time and was found in a majority of *H. zea* populations [10]. Furthermore, this contrasts with earlier estimates of Cry1Ac resistance in this region, where it was detected in 57% of the populations tested during 2016 [5] and where extensive resistance to Cry1A.105 and Cry2Ab2 was detected in North Carolina and South Carolina during 2017 and 2018 [17]. Our study highlights the variability in resistance that can be detected in populations across the region.

In a similar study, where *H. zea* larvae were also collected from various plant hosts, higher survival in bioassays was correlated with increased injury to Bt cotton in the field. In this mid-southern USA study, for every 1% increase in survival on a diagnostic dose in a bioassay (10 μg/cm^2^ toxin of diet for both Cry1Ac and Cry2Ab2), Bt cotton square and boll damage increases by 0.5% [10]. Based on our bioassays and this information, Bt cotton in some locations across the southeastern USA is at risk for *H. zea* infestation as a result of resistance. It is also important to note that many of the colonies we collected did not produce enough viable offspring to run a bioassay. Fitness costs and sublethal effects due to Bt resistance are well-documented in laboratory-selected and field-collected populations of *H. zea* [24,25,26,27,28,29]. A recent study found that several Cry-resistant populations of *H. zea* collected from the mid-southern USA lacked fitness costs and marked sub-lethal effects [30]. However, this is not surprising given the high levels of Cry resistance present in those populations, and the fact that sublethal effects due to Bt resistance decrease as resistance evolution increases. Another study in North Carolina and South Carolina found that impacts on pupal weight for *H. zea* collected from corn expressing Cry1A.105 + Cry2Ab2 had fallen to close to zero by 2019 [29]. Therefore, it is possible that populations we collected that did not produce enough viable eggs for a bioassay were experiencing fitness costs from Bt resistance. In that case, our observation that only 37.5% and 62.5% of the tested populations were resistant to Cry1Ac and Cry2Ab, respectively, underestimates the true extent of Bt resistance present in the southeastern USA landscape.

Previous studies have reported percentage mortality, weight and larval inhibition values at the highest dose tested in their bioassay [5,31], but have not tested the relationship between these values and LC_50_ values. We found that percentage mortality values at the highest dose were negatively related with LC_50_ values, which should not be surprising because the LC_50_ values were calculated on survival estimates in the bioassays. Furthermore, diagnostic dose tests are widely accepted as informative in resistance monitoring, and even preferable when resistance alleles are rare for F_2_ screens [32]. The only other significant regression was between LC_50_ values and larval weight inhibition with Cry2Ab2. We likely did not see relationships between LC_50_ values and larval instar inhibition at the highest dose, because we used the molting inhibitory concentration for our calculations (only counting larvae that molted to the second instar), and many of the larvae at the end of the bioassay were second or third instars after this exclusion. This meant that there was not a wide range of instar sizes and, therefore, we did not expect to see relationships with LC_50_ values.

We found that resistance rations were lower for Cry1Ac (15) than for Cry2Ab2 (53), which might suggest that there is less Cry1Ac resistance than Cry2Ab2 resistance in the southeastern USA. In contrast, a related study with a much larger sample size (95 *H. zea* populations from the mid-southern USA and Texas) found that *H. zea* survival was higher on a diagnostic dose of Cry1Ac compared to Cry2Ab2 [10]. However, our comparison only included nine populations from 2019. Therefore, we cannot make strong conclusions concerning the relative frequency of resistance to these two toxins in the southeastern USA.

Our study demonstrates the importance of continued resistance monitoring and the potential for Cry resistance ratios for *H. zea* to fluctuate in the environment. Resistance levels of *H*. *zea* to Cry1Ac, and larval damage to corn expressing similar Cry toxins, are highly dependent on crop composition in the local landscape [13,14]. Furthermore, the dominance of Cry2Ab2 resistance in this pest varies across toxin dose and source [33]. Therefore, it is not surprising that Bt resistance levels will vary among years, as environmental conditions shift. However, the lack of predictability makes management of this pest in Bt crops challenging. For example, in cotton, the most effective management method is to time foliar applications of chlorantraniliprole before *H. zea* larvae are established in the crop [6,34]. Furthermore, this insect can cause 100% yield loss in non-Bt cotton if it is not controlled [35]. Because of this, pest management practitioners must assume that all populations are Cry-resistant, since the risk of assuming a population is not resistant when it actually is, is too great. Understanding how the environment influences population abundances, and the frequency of resistance in this pest, will be key for future predictability.

In conclusion, our finding that the frequency of populations resistant to Cry1Ac Cry2Ab2 was variable (37.5% and 62.5% of the tested populations were resistant to Cry1Ac and Cry2Ab, respectively), provides an important contribution to previous studies, both in the southeastern USA and outside the region. Selection pressure remained in the environment during the study, as nearly all corn planted in the region expressed Cry1A, a majority expressed Cry2A [13], and a majority of cotton expressed both Cry1A and Cry2A [36,37]. As a result, we were surprised that the percentages of resistant populations and their resistance levels, relative to other studies, were not even higher. Many other environmental variables are likely important for driving Cry resistance in populations of *H. zea* beyond the extent of Bt crop plantings across the region and time since the first detection of resistance in the population.

## Figures and Tables

**Table 1 insects-14-00262-t001:** Bioassay results for Cry1Ac, 2018.

Location	LC_50_ μg/cm^2^	95% CI μg/cm^2^	Wald *Χ*^2^ Value	Slope	*p* Value	Resistance Ratio Relative to Benzon	Resistance Ratio Relative to Susceptible	Percentage Mortality at 10 μg/cm^2^	% Larval Inhibition (Weight) at 10 μg/cm^2^	% Larval Inhibition (Instar) at 10 μg/cm^2^
Benzon	0.32	0.18, 0.58	49.86	1.73	<0.0001			100		
Susceptible	0.33	0.07, 0.72	9.48	1.69	<0.0001			100		
Sumter Co., GA	0.21	0.10, 0.44	39.26	0.59	<0.0001	1	1	94	39	13
Lenoir Co., NC	0.62	0.36, 1.15	56.22	0.77	<0.0001	2	2	97	90	30
Sampson Co., NC	5.89	2.72, >10	38.31	0.70	<0.001	18	18	81	89	31
Barnwell Co., SC	>10	5.44, >10	8.55	0.43	0.0035	>30	>30	50	71	17
Darlington Co., SC	0.01	<0.01, 0.01	19.01	1.16	<0.0001	<1	<1	100		

**Table 2 insects-14-00262-t002:** Bioassay results for Cry1Ac, 2019. 1 and 2 refer to different collections from the same county, but different locations within the county.

Location	LC_50_ μg/cm^2^	95% CI μg/cm^2^	Wald *Χ*^2^ Value	Slope	*p* Value	Resistance Ratio Relative to Benzon	Resistance Ratio Relative to Susceptible	Percentage Mortality at 10 μg/cm^2^	% Larval Inhibition (Weight) at 10 μg/cm^2^	% Larval Inhibition (Instar) at 10 μg/cm^2^
Benzon	0.27	0.19, 0.35	59.91	3.14	<0.0001			90	99	49
Susceptible	0.55	0.42, 0.69	83.73	3.03	<0.0001			97	>99	43
Henry Co., AL	>10	>10, >10	5.90	1.27	0.0151	>37	>18	14	93	41
Santa Rosa Co., FL 1	>10	>10, >10	35.16	1.62	<0.0001	>37	>18	33	94	41
Washington Co., NC 1	6.96	2.34, >10	7.45	1.54	0.0063	26	13	72	93	35
Washington Co., NC 2	1.11	0.09, 3.94	11.16	0.94	0.0008	4	2	83	97	33
Edgecombe Co., NC	2.38	0.54, >10	15.61	0.66	<0.0001	9	4	73	86	30
Surry Co., NC	0.34	0.17, 0.57	53.05	1.64	<0.0001	1	<1	91	97	38
Wayne Co., NC	0.77	0.03, 1.50	7.22	2.87	0.0072	3	1	100		
Wilkes Co., NC	7.03	4.25, >10	11.15	3.24	0.0008	26	13	73	85	27
Barnwell Co., SC 1	1.01	0.45, 2.42	38.04	0.75	<0.0001	4	2	84	79	34
Barnwell Co., SC 2	3.89	- ^a^	3.77	0.19	0.0523	14	7	52	92	36
Florence, SC	0.62	0.28, 1.13	37.64	1.20	<0.0001	2	1	81	98	34

^a^ Not estimable.

**Table 3 insects-14-00262-t003:** Bioassay results for Cry2Ab2, 2019. 1 and 2 refer to different collections from the same county, but different locations within the county.

Location	LC_50_ μg/cm^2^	95% CI μg/cm^2^	Wald *Χ*^2^ Value	Slope	*p* Value	Resistance Ratio	Percentage Mortality at 10 μg/cm^2^	% Larval Inhibition (Weight) at 10 μg/cm^2^	% Larval Inhibition (Instar) at 10 μg/cm^2^
Benzon	0.07	0.03, 0.12	69.85	1.91	<0.0001		100		
Henry Co., AL	5.04	3.28, 7.93	22.82	2.14	<0.0001	72	75	95	45
Santa Rosa Co., FL 1	>10	14.86, >10	10.99	1.71	0.0009	>143	28	95	46
Washington Co., NC 1	3.69	1.66, 6.62	12.46	1.67	0.0004	53	75	97	47
Washington Co., NC 2	0.19	<0.1, 0.74	9.92	1.30	0.0016	3	97	71	4
Edgecombe Co., NC	1.12	0.23, 3.10	17.10	1.04	<0.0001	16	78	91	26
Surry Co., NC	1.38	<0.1, 5.60	5.98	1.29	0.0144	20	92	91	31
Wayne Co., NC	0.08	0.01, 0.27	23.63	1.28	<0.0001	1	97	79	17
Wilkes Co., NC	2.52	1.16, 6.96	55.74	0.88	<0.0001	36	36	94	18
Barnwell Co., SC 1	9.06	6.92, 13.21	16.00	4.09	<0.0001	129	58	97	41
Barnwell Co., SC 2	1.24	0.69, 2.42	56.50	0.96	<0.0001	18	69	94	38
Florence, SC	0.14	0.05, 0.46	39.90	−1.31	<0.0001	2	95	91	29
Suffolk, VA	>10	8.36, >10	22.19	2.52	<0.0001	>143	47	95	42

## Data Availability

Data are available upon request.

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
