# Peer review of "Magnitude and Extent of Helicoverpa zea Resistance Levels to Cry1Ac and Cry2Ab2 across the Southeastern USA"

_insects, 2023, doi:10.3390/insects14030262_

Round 1

Reviewer 1 Report (Previous Reviewer 1)

The authors have revised the manuscript according to my suggestions and answered these questions point by point. Im satisfied with all revisions they have made. However, two spelling errors including Cry2Ab in Line 34 and CryAb2 in Line 35 should be corrected as Cry2Ab2. So I think this manuscript is suitable for publication at present.  

Author Response

The authors have revised the manuscript according to my suggestions and answered these questions point by point. I’m satisfied with all revisions they have made. However, two spelling errors including “Cry2Ab” in Line 34 and “CryAb2” in Line 35 should be corrected as “Cry2Ab2”. So I think this manuscript is suitable for publication at present.  

Change made

Reviewer 2 Report (Previous Reviewer 3)

The corrections are properly processed.

Author Response

The corrections are properly processed

Thank you

Reviewer 3 Report (New Reviewer)

Reisig et al. investigated the level of resistance of field-collected strains of Helicoverpa zea to two major Bt toxins, Cry1Ac and Cry2Ab2, expressed in Bt cotton and other transgenic crops. Almost 30 samples have been collected across the southeastern USA in 2018 and 2019, but several field strains failed to develop and did not lay enough eggs, so that only a smaller fraction was tested. Neonates from field populations were tested in diet overlay bioassays in comparison to two susceptible reference strains. The obtained data revealed the presence of resistance alleles against both toxins and in most populations tested, rendering the findings extremely useful to inform future pest management decisions. The paper is well-written, and the methods are described in enough detail. I only have a few minor points for the authors to consider when asked to submit a revised version of their paper.

The discussion is quite verbose in some places, especially those parts discussing the LC50 values obtained for the susceptible reference strains. I wonder why the authors have not calculated all resistance ratios based on the response of the Benzon strain. This strain is well accepted as a susceptible laboratory reference strain and obviously provided more stable results when compared to the variable results obtained for the susceptible lab strain maintained by the authors. That said, I would suggest for data harmonization to calculate the resistance ratios for both toxins tested based on the response of the Benzon strain.

I think it would be good to give the full name of Bt when first mentioned, because not all readers may know what Bt means.

L34: Insert “resistant to”

L126: Please explain which buffer you used.

All tables: Please print the 95% confidence intervals next to the calculated LC50.

Table 3: According to the methods section the highest concentration tested was 10 microgr/cm2, so I wonder about the LC50s given for the Santa Rosa and Suffolk strain, respectively. Are these values extrapolated? This is in contrast to table 1 and table 2 (>10 microgr/cm2).

Author Response

Reisig et al. investigated the level of resistance of field-collected strains of Helicoverpa zea to two major Bt toxins, Cry1Ac and Cry2Ab2, expressed in Bt cotton and other transgenic crops. Almost 30 samples have been collected across the southeastern USA in 2018 and 2019, but several field strains failed to develop and did not lay enough eggs, so that only a smaller fraction was tested. Neonates from field populations were tested in diet overlay bioassays in comparison to two susceptible reference strains. The obtained data revealed the presence of resistance alleles against both toxins and in most populations tested, rendering the findings extremely useful to inform future pest management decisions. The paper is well-written, and the methods are described in enough detail. I only have a few minor points for the authors to consider when asked to submit a revised version of their paper.

The discussion is quite verbose in some places, especially those parts discussing the LC50 values obtained for the susceptible reference strains. I wonder why the authors have not calculated all resistance ratios based on the response of the Benzon strain. This strain is well accepted as a susceptible laboratory reference strain and obviously provided more stable results when compared to the variable results obtained for the susceptible lab strain maintained by the authors. That said, I would suggest for data harmonization to calculate the resistance ratios for both toxins tested based on the response of the Benzon strain.

Excellent point and we have added this to the tables where possible

I think it would be good to give the full name of Bt when first mentioned, because not all readers may know what Bt means.

Change made

L34: Insert “resistant to”

Change made

L126: Please explain which buffer you used.

We added this detail

All tables: Please print the 95% confidence intervals next to the calculated LC50.

Change made

Table 3: According to the methods section the highest concentration tested was 10 microgr/cm2, so I wonder about the LC50s given for the Santa Rosa and Suffolk strain, respectively. Are these values extrapolated? This is in contrast to table 1 and table 2 (>10 microgr/cm2).

Thank you for pointing this out and we corrected these

This manuscript is a resubmission of an earlier submission. The following is a list of the peer review reports and author responses from that submission.

Round 1

Reviewer 1 Report

Totally, the manuscript was well organized and easy to follow. The methods, results and conclusions are scientifically sound. Their findings will greatly contribute to resistance management of Helicoverpa zea in the genetically modified crops expressing insecticidal Cryproteins from Bacillus thuringiensis. However, this manuscript has few of errors as details in the lists of major and minor concerns as currently written. So I don’t think this manuscript is suitable for publication at present.

There are some deficiencies as illustrated in the following:

Major concerns: 

1. In the end of Introduction, the authors should added a sentence to introduce the purpose and significance of this study.

2. In the section of Materials and Methods, I strongly recommended the subtitles should be used in different sections, such as “2.1. Insect collections and colonies” (in front of Line 88), “2.2. Bioassays” (in front of Line 104), and “2.3. Statistical analyses” (in front of Line 138).

3. In tables 1-3, the unit of LC50 and 95% CI should be added. In addition, what did the authors want to express using “P > X2” ? I think it may be “P value” only.

Minor concerns:

1. In Line 16, effectively manage may be better than best manage .

2. In Line 26, the word of cotton should be added in the end of Cry-expressing.

3. In Line 28, monitored should be revised as monitored for.

4. In line 35, to should be added in the end of resistant.

5. In Line 42, resistance should be listed in the section of Keywords.

6. In Line 93, 4M should be corrected as 4 M. In addition, Vanderzant Vitamin Mixture should be revised as Vanderzant vitamin mixture.

7. In Line 102, the manufacturer information and its key technical parameters of a growth chamber should be added.

8. In Line 141, report should be changed into reported.

9. In Line 155, É‘ < 0.05 should be revised as P < 0.05.

10. In Lines 169-170, the sentence should be corrected as follows:

Resistance ratios were calculated based on the LC50 value of the Benzon colony as the susceptible colony and ranged from 2 (Florence, SC) to over 451 (Santa Rosa Co., FL).

Finally, I hope the authors can use these to correct the same problem for the rest.

Reviewer 2 Report

Bt resistance is an increasing developed problem in H.zea. This study made broad investigations by evaluating the extent and magnitude of Cry1Ac resistance in H.zea across the southeastern USAduring 2018 and 2019 and Cry2Ab2 resistance during 2019.
Here are three questions:
    1.Why did not perform Cry2Ab2 assays in 2018?

    2.It has been reported that many other insects developed high levels of Cry toxin resistance. The comparison with other insects report should be made in discussion.

    3.Why do H.zea show the different resistance to Cry1Ac and Cry2Ab2 in your investigation,could you make analysis in discussion?

Reviewer 3 Report

This paper reports on the insecticidal resistance of pests that can damage insect-resistant crops. The impact of the data itself is not that great, but it contains valuable information for promoting future research in this field. I will accept it as it is because the point of the argument is solid.